# Computational screening methodology identifies effective solvents for $CO_2$ capture

Alexey A. Orlov[1], Alain Valtz[2], Christophe Coquelet[2], Xavier Rozanska[3], Erich Wimmer[3], Gilles Marcou[1], Dragos Horvath[1], Bénédicte Poulain[4], Alexandre Varnek [1✉] & Frédérick de Meyer [2,4✉]

Carbon capture and storage technologies are projected to increasingly contribute to cleaner energy transitions by significantly reducing $CO_2$ emissions from fossil fuel-driven power and industrial plants. The industry standard technology for $CO_2$ capture is chemical absorption with aqueous alkanolamines, which are often being mixed with an activator, piperazine, to increase the overall $CO_2$ absorption rate. Inefficiency of the process due to the parasitic energy required for thermal regeneration of the solvent drives the search for new tertiary amines with better kinetics. Improving the efficiency of experimental screening using computational tools is challenging due to the complex nature of chemical absorption. We have developed a novel computational approach that combines kinetic experiments, molecular simulations and machine learning for the in silico screening of hundreds of prospective candidates and identify a class of tertiary amines that absorbs $CO_2$ faster than a typical commercial solvent when mixed with piperazine, which was confirmed experimentally.

[1] Laboratory of Chemoinformatics, Faculty of Chemistry, University of Strasbourg, 67081 Strasbourg, France. [2] MINES ParisTech, PSL University, Centre of Thermodynamics of Processes (CTP), 35 rue St Honoré, 77300 Fontainebleau, France. [3] Materials Design SARL, 42 avenue Verdier, 92120 Montrouge, France. [4] TOTALEnergies S.E., OneTech, Gas & Low Carbon Entity, CCUS R&D Program, 2 Place Jean Millier, 92078 Paris, France. ✉email: varnek@unistra.fr; frederick.de-meyer@totalenergies.com

Numerous technologies exist for capturing $CO_2$ including chemical absorption, cryogenic separation, removal with membranes, and adsorption with zeolites or metal–organic frameworks[1–6]. The cyclic chemical absorption and regeneration process based on common primary and secondary amines such as monoethanolamine (MEA) and diethanolamine (DEA) is the most mature in industrial applications[3,5]. Unhindered primary and secondary amines react rapidly with $CO_2$ to form very stable carbamates. The amount of energy required for the regeneration of these solvents is large. Carbon capture applied to a coal-fired power plant may reduce the net output of the plant by 30%[6]. With sterically hindered amines or tertiary amines like the standard methyldiethanolamine (MDEA), $CO_2$ is captured as bicarbonate, which has a much smaller heat of reaction than carbamate formation, resulting in regeneration energy savings[7]. Moreover, their $CO_2$ absorption capacity is much higher. Tertiary amines are therefore increasingly used in the high-pressure natural gas treatment industry to remove acid gases like $CO_2$. However, in general, the rate of direct bicarbonate formation is much lower than that of carbamate formation resulting in much slower $CO_2$ absorption rates with tertiary amines and thus in unacceptable large equipment for low pressure, anthropogenic (flue gas), $CO_2$ capture applications[5,7]. To tackle this problem, several approaches were suggested. Several studies reported that the usage of a catalyst allows one to speed up the absorption of $CO_2$ and/or to lower the energetic cost of solvent regeneration[8]. Another option, which is currently followed by the industry, consists in adding an activator, piperazine, significantly boosting the overall $CO_2$ absorption rate without increasing the regeneration energy too much[9]. A more straightforward strategy would be the identification of new tertiary amines with much higher absorption rates with respect to standard MDEA and to which piperazine can eventually be added. Since experimental measurement of $CO_2$ absorption kinetics is a time and labor-intensive process, the rational approach to the design of tertiary amines that can rapidly absorb $CO_2$ requires a quantitative model enabling to select only the best candidates for experimental measurements.

Concerning alternative processes based on adsorption in porous solids (still under development), a lower theoretical energy consumption is expected due to the weaker physical adsorption. Molecular simulations and machine learning have already been extensively used to perform virtual screening of hundreds of thousands of structures to identify potentially better materials for $CO_2$ adsorption[10,11]. Until now it was not possible to apply a similar methodology for amines, because of the difficulty related to the computation of chemical reactions. Amines were rationally designed based on physical and thermodynamic properties and the $CO_2$ absorption rates were measured experimentally for only the most promising candidates[7,12]. Previously, machine-learning algorithms were tentatively applied for modeling quantitative structure–property relationship (QSPR) of alkanolamines' $CO_2$ absorption-related properties[13–18]. However, the availability of only a very small amount of data points limited the applicability domain of the models. Hence, to address this issue, we developed and applied a methodology for the identification of tertiary amines effectively absorbing $CO_2$ based on the combination of molecular simulations[19] and machine learning. In parallel, an experimental setup for the measurement of $CO_2$ absorption rates has been specifically designed and put in place to validate the approach.

## Results and discussion
### Design of the methodology for $CO_2$ absorbents screening.
The workflow of the methodology is presented in Fig. 1. Chowdhury et al.[20] published a consistent experimental dataset of the absorption rates of $CO_2$ for 24 aqueous tertiary amines (313 K, 30 wt% amine). In the absence of a clear relationship between the structure or the chemical properties (e.g., the basicity) of the amines and the $CO_2$ absorption rates, we developed a molecular dynamics (MD) based model that can accurately predict those experimental $CO_2$ absorption rates[19]. It was found that, while the basicity of the amine (quantified by the $pK_a$) is important, the key to the precision of molecular simulations is the inclusion of subtle but important solvation effects in the calculation of the activation Gibbs free energy of the reaction with an accuracy better than 1 kJ mol$^{-1}$. One of the important features of the MD model[19] is the robustness to reasonable changes in the concentration of amine and in temperature, enabling to apply it to a rather wide range of experimental setups. Hence, the model was applied to predict the rates at 13 mol% of amines and at 323 K, because these conditions are more representative of industrial absorption[5].

Being much less resource- and cost-demanding, molecular simulations can thus be used instead of the experiments to get enough data for building a reliable QSPR model with a wide applicability domain.

### Molecular simulations of $CO_2$ absorption process.
A dataset containing 100 structurally diverse tertiary amines was composed based on the in-house TotalEnergies's dataset of amines with known experimental properties, complemented with tertiary amines extracted from literature and public databases (PubChem[21,22], ZINC[23,24]). The selected compounds comprise diverse chemotypes, including linear and cyclic amines, diamines, amines containing thiol and thioether groups. Molecular simulations (see "Methods") were performed for the initial set of 24 amines and for the selected set of 100 amines at 323 K and using a 13 mol% concentration of amine. From MD simulations absorption rates ($R_{MD}$) and free energies of absorption ($\Delta G_{MD}$) were obtained. Notably, the $R_{MD}$ values calculated at 313 and 323 K are highly correlated (Fig. 2a, Spearman rank correlation coefficient ($\rho$) 0.99).

As shown in Fig. 2b, the most rapidly absorbing compound according to the MD calculations and the data from Chowdhury et al.[20] was 3-(Diethylamino)-1,2-propanediol (DEA-1,2-PD). However, most of the other compounds with the largest predicted rates of absorption ($R_{MD}$) contained either piperidine or pyrrolidine cycles. This is in line with the data from Chowdhury et al.[20], who showed that 3-piperidino-1,2-propanediol (3PP-1,2-PD) and 1-methyl-2-piperidineethanol were significantly faster than the industrially used methyldiethanolamine (MDEA). Figure 2c illustrates that the computed $CO_2$ absorption Gibbs free energies $\Delta G_{MD}$ are almost perfectly correlated with the $CO_2$ absorption rates, $R_{MD}$ (Spearman $\rho$ −0.98): the slower the $CO_2$ absorption, the higher the absorption Gibbs free energy. The correlation is not linear, and the decrease of $\Delta G_{MD}$ slows down significantly at higher $CO_2$ absorption rates.

### Virtual screening of tertiary amines and experimental validation.
Machine-learning algorithms were applied to establish quantitative structure–property relationships and screen a set of tertiary amines from a public dataset. The values of $pK_a$ predicted by the OPERA model[25] can be used as a rather good predictor for $\Delta G_{MD}$. Indeed, the fitting of linear regression with the $pK_a$ values as the only predictor leads to a reasonable predictive performance in cross-validation (Supplementary Table 2). For modeling both end-points ($\Delta G_{MD}$ and $R_{MD}$), we implemented a machine-learning workflow combining several machine-learning algorithms and various descriptors of molecular structures. Thus, predicted $pK_a$ values were complemented with other descriptor types: physicochemical descriptors from OPERA and various types of molecular fragments calculated using ISIDA-Fragmentor[26,27]. Finally, we used a consensus of several individual models built with the help of random forest (RF)[28] and

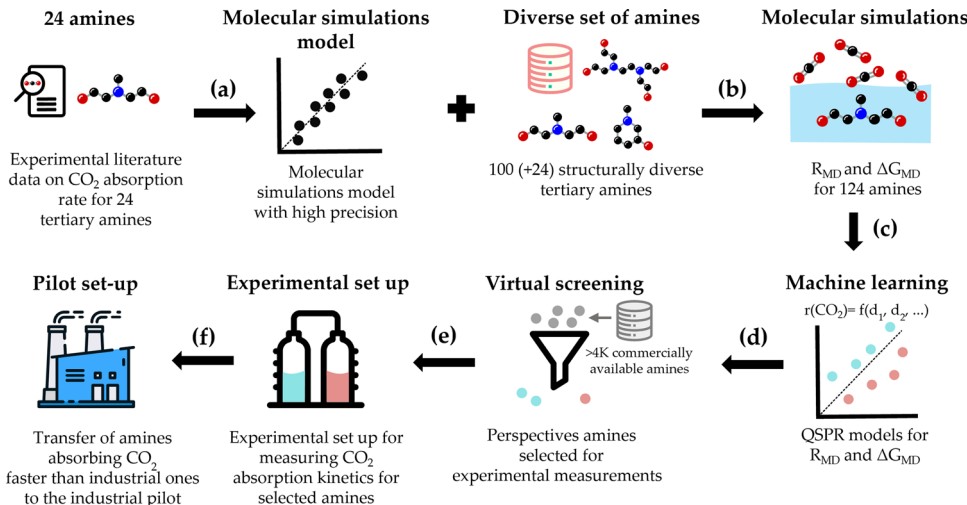

**Fig. 1 Workflow of the methodology suggested in this paper. a** A high precision molecular simulation-based model for absorption rate prediction is developed[19] and validated with experimental data on $CO_2$ absorption rates for 24 tertiary amines[20]. The accuracy of the Gibbs free energies of absorption is better than 1 kJ mol$^{-1}$ in comparison to experimental values[19]. **b** This model is applied to a diverse dataset containing 100 tertiary amine structures to calculate the $CO_2$ absorption rate ($R_{MD}$) and free energy of absorption ($\Delta G_{MD}$) (see "Methods"). **c** QSPR models were built for $R_{MD}$ and $\Delta G_{MD}$. **d** QSPR models were used to select perspective commercially available amines from public datasets. **e** Experimental measurement of $CO_2$ absorption rates for selected amines. **f** The most selective ones can be further studied and eventually tested in a pilot unit.

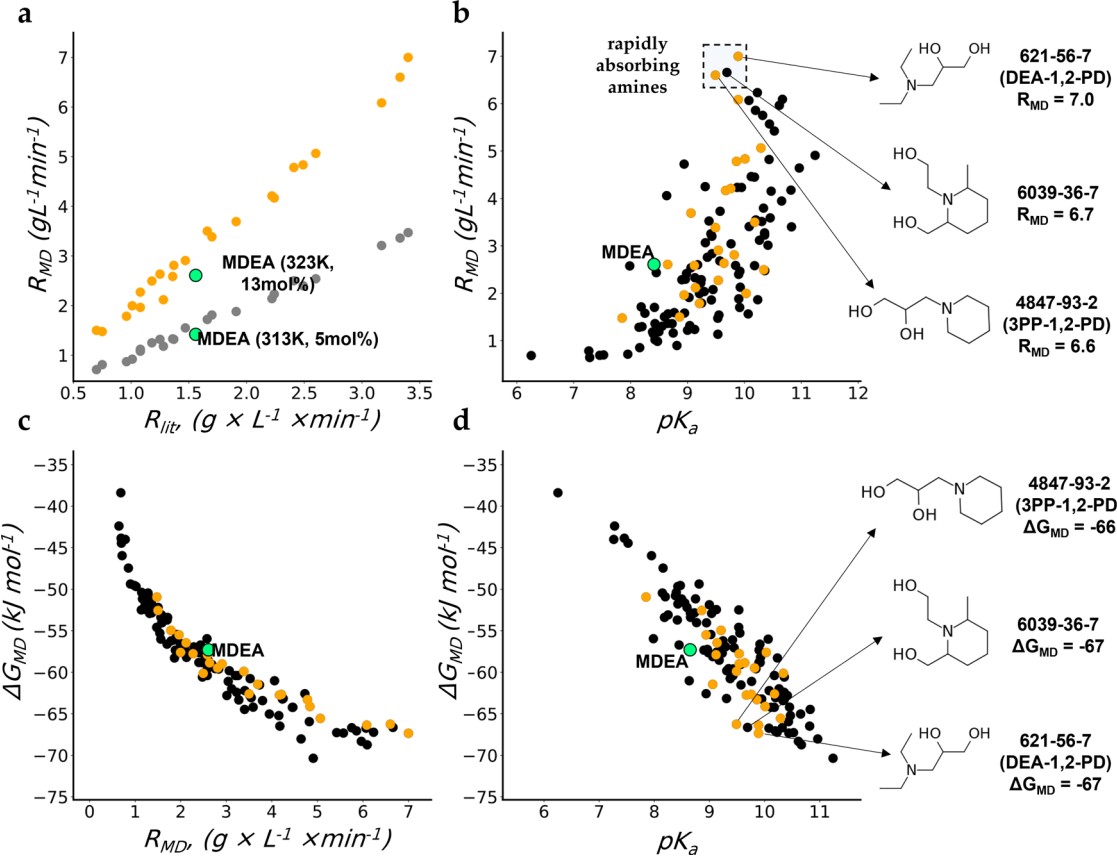

**Fig. 2 Results of molecular dynamics simulations of the $CO_2$ absorption process. a** $CO_2$ absorption rates ($R_{MD}$) at 313 K (gray) and 323 K (orange) predicted using MD and the experimental absorption rates ($R_{lit}$) at 313 K. **b** $R_{MD}$ vs predicted $pK_a$ values ($pK_a$). **c** energy of absorption ($\Delta G_{MD}$) predicted by MD vs $R_{MD}$. **d** $\Delta G_{MD}$ vs predicted $pK_a$. The 24 amines from Chowdhury et al.[20] are shown in orange. The 100 amines for which MD simulations were performed are shown in black. Industrially used reference compound (MDEA) is shown in green.

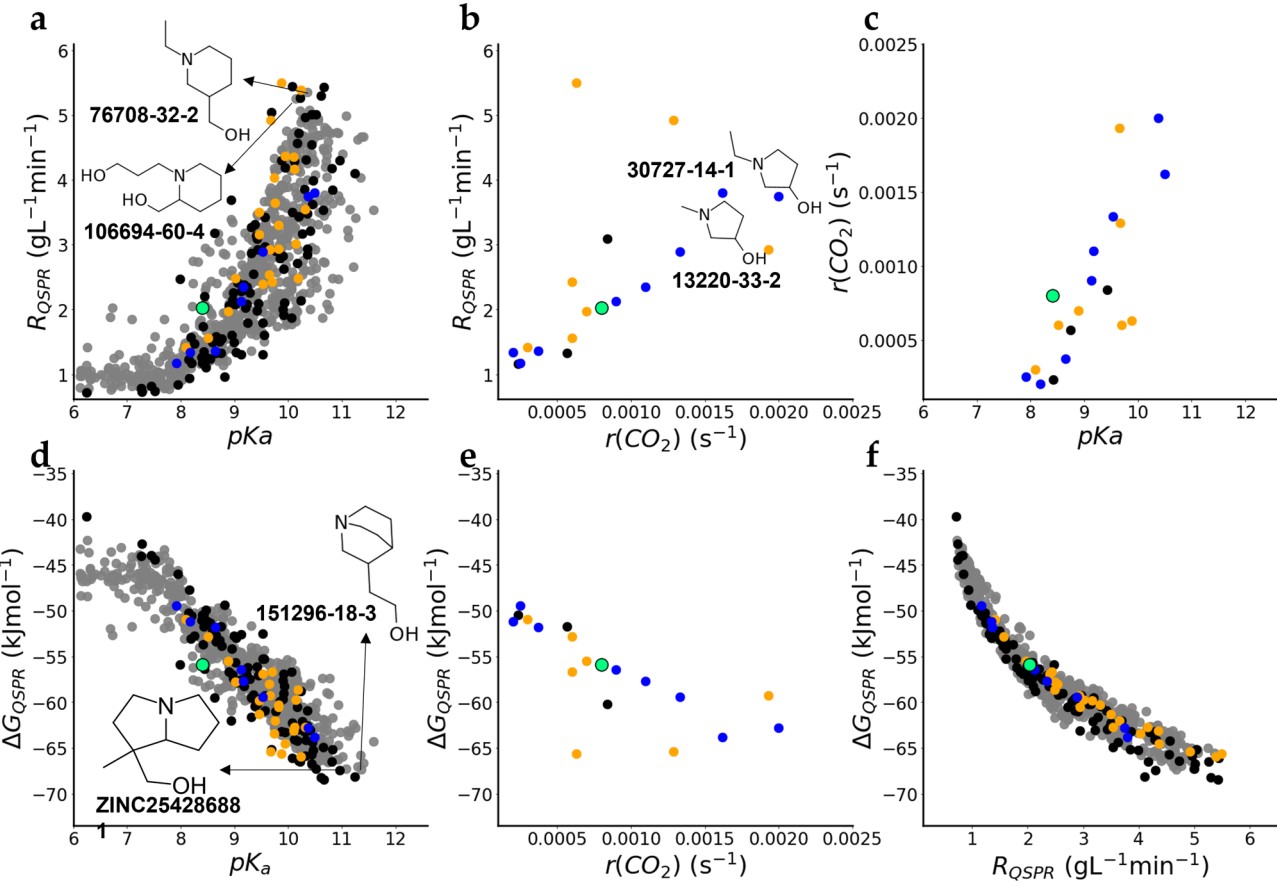

**Fig. 3 Virtual screening of tertiary amines and experimental validation. a** Absorption rates predicted by the QSPR model ($R_{QSPR}$) vs predicted $pK_a$ values. **b** $R_{QSPR}$ vs experimentally measured absorption rate ($r(CO_2)$). **c** $r(CO_2)$ vs predicted $pK_a$. **d** Free energies of absorption predicted by QSPR model ($\Delta G_{QSPR}$) vs predicted $pK_a$. **e** $\Delta G_{QSPR}$ vs $r(CO_2)$. **f** $\Delta G_{QSPR}$ vs $R_{QSPR}$. Amines present in the initial dataset from Chowdhury et al.[19,20] are shown in orange. Amines selected for MD simulations in the present work are shown in black. The industrially used reference compound (MDEA) is shown in green. Eight novel amines which were not present in the training set are shown in blue. The CAS numbers of the most perspective compounds are shown.

eXtreme Gradient Boosting (XGBoost))[29] machine-learning algorithms on a merged subset of ISIDA fragments and descriptors generated with the OPERA tool. Although the predictive accuracy in terms of RMSE is of the same order of magnitude as in Kuenemann et al.[13] for absorption rates (Supplementary Table 2 and Supplementary Fig. 1), the applicability domain of our models is much larger, since the training set contained three times more compounds. It is worth noting that a QSPR model which did not allow one to achieve an excellent accuracy can still be useful for ranking the amines from the large compounds databases[13,30]. Therefore, we retrieved from the public database ZINC[23] the tertiary amines which were not too large (Mw ≤ 250 gmol⁻¹), not too lipophilic ($-1 \leq$ clogP $\leq 1$), and readily available from suppliers. In total, more than 800 amines were screened virtually. Numerous amines outranking MDEA in terms of the predicted absorption rates ($R_{QSPR}$) were identified (Fig. 3a). For example, various substituted piperidines were among the compounds with the largest $R_{QSPR}$ (Fig. 3a).

**Experimental measurement of the $CO_2$ absorption kinetics**. An experimental setup was put in place to measure and compare the rate of $CO_2$ absorption in aqueous tertiary amines. For each experiment, the same initial amount of $CO_2$ was set in contact with the solvent and the evolution toward equilibrium of the partial pressure of $CO_2$ in the gas phase was measured over time. The slope of the absorption curve at the time at which 50% of the

$CO_2$ was absorbed (with respect to the equilibrium value) was calculated ($r(CO_2)$). It is a measure of the rate of $CO_2$ absorption. Eighteen amines comprising 7 amines from the initial set of 24 amines from Chowdhury et al.[20], 3 amines from the diverse dataset of 100 amines, and 8 novel amines that were never present in the training set were purchased and an assessment of their absorption rate was performed (Fig. 3b, c, e and Supplementary Tables 3 and 4). Both $\Delta G_{QSPR}$ and absorption rates $R_{QSPR}$ were highly correlated with $r(CO_2)$ for eight novel amines (Spearman ρ 0.93) as well as the predicted $pK_a$ values. Five out of eight purchased amines absorbed $CO_2$ faster than MDEA. Two amines: 1-methyl- and 1-ethyl-3-pyrrolidinol (EPOL) were especially effective. These compounds represent an interesting class of the tertiary amines, which to our knowledge have not been explored yet.

While tertiary amines like the standard MDEA are often used for high-pressure natural gas treatment, they are not suitable for low-pressure anthropogenic $CO_2$ removal due to the low $CO_2$ absorption rate. Activators such as piperazine can be added to enhance the $CO_2$ absorption rate. The impact of piperazine is shown in Fig. 4 for two amines, namely MDEA and EPOL. The latter is a tertiary amine that has been selected for its fast $CO_2$ absorption rate following the virtual screening. In the absence of piperazine EPOL absorbs $CO_2$ much faster than MDEA. The addition of piperazine significantly enhances the $CO_2$ absorption rates with EPOL + PZ showing the fastest absorption.

In conclusion, a methodology for computer-aided design of tertiary amines effectively absorbing $CO_2$ was suggested in this

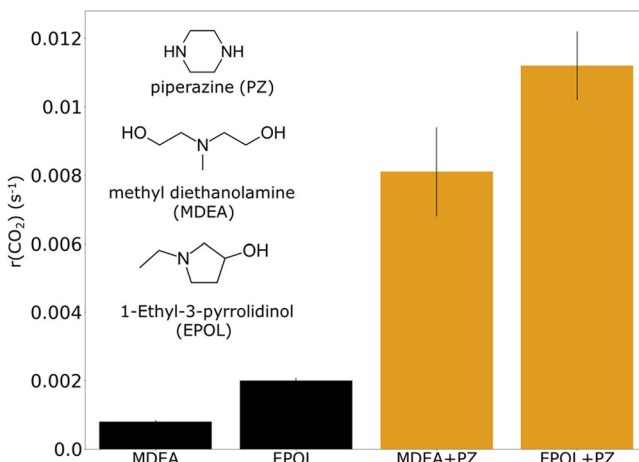

**Fig. 4 Experimental kinetic measurements with piperazine.**
Experimentally measured $CO_2$ absorption rate ($r(CO_2)$) of standard MDEA and EPOL, a new amine suggested in this work, and their mixtures with piperazine (+PZ). Aqueous alkanolamine mixtures contain 13 mol% amine and water and mixtures with PZ contain 11 mol% amine, 2.5 mol% PZ and water. Standard deviations of the values are shown as error bars.

paper. The methodology is based on the combination of state-of-the-art molecular dynamics simulations that generate a sufficiently large dataset that are used as an input for machine-learning modelling followed by large-scale virtual screening. In parallel, the approach is experimentally validated. It allowed the identification of amines that absorb $CO_2$ faster than those currently used in the industry. Since the development of an optimal solvent is a multi-objective task, we believe that the proposed methodology can be provisionally repurposed to application for modeling of other industrially important properties of alkanolamine-based solvents.

## Methods

**Molecular simulations**. The approach developed recently and described in Rozanska et al.[19], was used to compute the rates of $CO_2$ absorption in aqueous amine solvents (see Supplementary Methods), which relies primarily on the solvation properties of $OH^-$, $CO_2$, and $HCO_3^-$. In this model, the tertiary amine solely acts as a base.

$$R_{MD} = A(T) \times \exp\left(\frac{-\Delta G^{\neq}}{RT}\right) \times [CO_2][OH^-] \quad (1)$$

The rates are obtained from Eq. (1) where $R_{MD}$ is the absorption rate, $[CO_2]$ and $[OH^-]$ are the concentrations of carbon dioxide molecules and hydroxyl anions, respectively, $\Delta G^{\ddagger}$ is the Gibbs free energy barrier for the reaction $CO_2 + OH^-$ to $HCO_3^-$, $RT$ is the macroscopic thermodynamic energy unit, where $R$ is the universal gas constant and $T$ the absolute temperature, and $A(T)$ is a temperature-dependent pre-exponential factor. In Eq. (1), $\Delta G^{\ddagger}$ is obtained from a Polanyi–Evans relation with as input the energy differences of solvation of $OH^- + CO_2$ (reactants) and $HCO_3^-$ (product) computed in the 124 aqueous amine solvents. The concentrations $[CO_2][OH^-]$ are obtained numerically solving pH equations, and $A(T)$ is fitted using the experimental rates of the reaction $CO_2 + OH^-$ in ten aqueous amine solvents. The Polanyi–Evans relation between $\Delta G^{\ddagger}$ and energy differences of solvation, $\Delta G$, of $OH^- + CO_2$ and $HCO_3^-$ is given by Eq. (2).

$$\Delta G^{\ddagger} = a\Delta G(T) + b \quad (2)$$

where $a$ and $b$ are fitted to reproduce the experimental rates in pure water and ten aqueous amine solvents and $\Delta G(T)$ is the energy difference of solvation of $OH^- + CO_2$ and $HCO_3^-$ obtained from molecular dynamics simulations. Additional details and the values for $A(T)$, $a$, and $b$ can be found in Rozanska et al.[19].

For the calculation of the regeneration energy, the following three reactions are considered:

$$OH^- + CO_2 = HCO_3^- \quad (3)$$

$$Amine + H_2O + CO_2 = ammonium + HCO_3^- \quad (4)$$

$$Amine + H_2O = ammonium + OH^- \quad (5)$$

The free energy of absorption is $\Delta G_4$ ($=\Delta G_{MD}$ in Fig. 2) $= \Delta G_3 + \Delta G_5$ with $\Delta G_3$

calculated from the molecular simulations ($\Delta G(T)$ in Eq. (2)) in every aqueous amine and $\Delta G_5$ calculated from the amine $pK_a$.

**Quantitative structure–property relationship modeling**. All compound structures were standardized using RDKit[31] nodes in KNIME[32]. The standardization procedure included aromatization, stereochemistry depletion, removal of salts/solvents, neutralization, removal of explicit hydrogens. Standardized structures for 124 amines are given in Supplementary Table 1 and at https://github.com/AxelRolov/CO2_chemical_solvents.

In all, 193 different ISIDA fragment descriptors were generated using the Fragmentor17 software[26,27]. These fragments represent either sequences (the shortest topological paths with an explicit presentation of all atoms and bonds), atom pairs, or triplets (all the possible combinations of three atoms in a graph with the topological distance between each pair indicated).

Various physicochemical properties ($pK_a$, logP, melting and boiling points, vapor pressure, water solubility, etc.) and several substructural fragments counts (ring count, heavy atom count, etc.) used as descriptors, were calculated using OPERA v.2.6[25].

All descriptors used in this work are available at https://github.com/AxelRolov/CO2_chemical_solvents.

Prior to the application of machine-learning algorithms $R_{MD}$ and $\Delta G_{MD}$ values were transformed to a logarithmic scale, i.e., the negative value of decimal logarithm was taken ($-\log_{10} R_{MD}$, $-\log_{10}(-\Delta G_{MD})$).

Random forest (RF): RF algorithm[28] implemented in sci-kit learn library (v. 0.22.1)[33], was used. The following hyperparameters were optimized (grid search): number of trees (100, 300, 1000), number of features (all features, one-third of all features, $\log_2$ of the number of features), the maximum depth of the tree (10, 30, full tree), bootstrapping (with and without the usage of bootstrap samples for building the tree).

XGBoost (XGB): XGBoost algorithm[29] as implemented in XGBoost python module (v.1.2.0; https://xgboost.readthedocs.io/en/latest/python/python_intro.html) was used. The following hyperparameters were tuned during optimization (grid search): number of trees (50, 100, 300, 500), number of features (all features, 70% of all features), number of samples (all samples, 70% of all samples), the maximum depth of the tree (5, 20, full tree), learning rate (0.3, 0.1, 0.5, 0.05). All other parameters were left as default.

Support vector regression (SVR): SVR algorithm[34] implemented in sci-kit learn library (v. 0.22.1), was used. The descriptors were scaled to the [0,1] range before applying the algorithm. The following hyperparameters were tuned during optimization (grid search): kernel (linear, rbf, poly, sigmoid), kernel coefficient (1, 0.1, 0.01, 0.001, 0.0001), regularization parameter (0.1, 1, 10, 100, 1000).

The modeling workflow was implemented using the sci-kit learn library (v. 0.22.1) in Python 3.7 scripting language (Supplementary Fig. 2). Identical modeling workflows were used for modeling absorption rates ($R_{MD}$) and energies of absorption ($\Delta G_{MD}$). The values were expressed as negative logarithms of base 10. At the first stage of the modeling, a machine-learning algorithm: RF, SVR, and XGB were tested in fivefold cross-validation, which was repeated five times. For each descriptor set, the model's measures of performance were calculated and several models with a coefficient of determination $Q^2_{CV} \geq 0.6$ for ($R_{MD}$) and $Q^2_{CV} \geq 0.7$ for ($\Delta G_{MD}$) were selected for consensus modeling. Consensus models were built for each descriptor type separately. In order to assess a propensity to predict data never seen during the training of the model, a nested cross-validation procedure[35] has been implemented. Here the method hyperparameters were found by optimizing the model performance in the fivefold cross-validation inner loop, while prediction was made for the test set from the outer loop, which represent a fold of the outer fivefold cross-validation cycle. To avoid a bias with the compounds numbering in the parent set, this procedure was repeated five times after reshuffling of the compounds. In such a way, the overall performance of the model ($Q^2_{NCV}$, $RMSE_{NCV}$, $MAE_{NCV}$) were estimated as an average of related statistical parameters obtained for each (out of 5) individual cross-validation loop.

Equations (6–8) were used to calculate the measures of the model's performance in cross-validation:

$$Q^2_{CV} = \frac{\sum_{j=1}^5 (1 - \frac{\sum_{i=1}^n (y_{i,exp} - y_{i,pred})^2}{\sum_{i=1}^n (y_{i,exp} - \bar{y})^2})}{5} \quad (6)$$

$$RMSE_{CV} = \frac{\sum_{j=1}^5 \sqrt{\sum_{i=1}^n \frac{(y_{i,exp} - y_{i,pred})^2}{n}}}{5} \quad (7)$$

$$MAE_{CV} = \frac{\sum_{j=1}^5 \sum_{i=1}^n \frac{|y_{i,exp} - y_{i,pred}|}{n}}{5} \quad (8)$$

Above, $n$ is the number of compounds in the learning set, $y_{i,exp}$, $y_{i,pred}$ experimental and values predicted in fivefold cross-validation for compound $i$ from the learning set, $j$ is the index of the repetition of the tenfold cross-validation procedure.

Each of the selected models was then associated with an Applicability Domain (AD), defined as a boundary box. The pool of selected models extracted from the given dataset can now be used as a consensus predictor, returning for each input solvent candidate a mean value of solubility estimates and its standard deviation,

taken over the predictions returned by each model in the pool or, alternatively, over the predictions returned by only those models having the candidate within their AD.

Outlying data points were defined as the data points, for which absolute errors ($|\chi exp-\chi pred|$) from cross-validation were larger than $2\times RMSE_{CV}$ threshold.

The absence of chance correlation was checked through the Y-randomization procedure. A Y-randomization test was performed in the following way: $-\log_{10}\chi$ values (y values) were shuffled, models were built using shuffled values and the values from the corresponding cross-validation test set were calculated. This procedure was repeated 100 times for each fold and the maximum values of the out-of-bag coefficient of determination were reported.

A library for virtual screening was performed in the following way. At first, all compounds from ZINC database with molecular weight no larger than 250 g/mol and calculated logP in the range of $(-1,1)$ were retrieved. Structures were standardized and then filtered. All compounds which did not contain tertiary amines, compounds, containing double bonds, aromatic rings, primary or secondary amine groups, ketones and sulfur-containing compounds except for thiols and thioethers were removed. Structures of screened compounds as well as predicted values are available at https://github.com/AxelRolov/CO2_chemical_solvents.

**Experimental measurement of $CO_2$ absorption rates**. To measure the kinetics of absorption and desorption of acid gases in aqueous amine solutions, a thermo-regulated constant interfacial area Lewis-type reactor cell was used[36]. The reactor (Supplementary Figs. 3–6) is equipped with an internal stirring system (magnetic stirrer) with the external motor. The operator needs to take care to select the speed of stirring without disturbing the interface (interface must be flat). Temperature is given by two platinum probes located at the upper and lower flanges (with the possibility to determine the gradient of temperature). The cell is immersed in a liquid bath. An electric resistor is introduced into the upper flange to control the gradient of temperature and avoid condensation of water and amine. Two capillary samplers are adapted to sample the vapor phase. The capillary samplers (ROLSI®, Armines' patent) are capable of withdrawing and sending micro samples to a gas chromatograph without perturbing the equilibrium conditions over numerous samplings, thus leading to repeatable and reliable results. Analytical work was carried out using a gas chromatograph (PERICHROM model PR2100, France) equipped with a thermal conductivity detector (TCD) connected to a data software system. Helium is used as the carrier gas in this experiment. The model of the GC column is Porapak R (Porapak R 80/100 mesh, 1 m × 2 mm ID Silcosteel). Each ROLSI® sampler is connected to a TCD. A tube allows either to evacuate or to introduce $CO_2$ from or into the cell. The kinetics of gas absorption are determined by recording the pressure drop through a calibrated pressure transducer. A computer equipped with data acquisition system records the pressure as a function of time.

The experimental procedure is the following:

The desired amount of solvent is introduced into the cell. The density obtained using a low-pressure vibrating tube densitometer (Anton Paar DSA 5000) is used to determine the exact mole number of solvent.

At least 5 bar of methane is added. We add methane because with this configuration, it is not possible to sample at pressures lower than GC carrier gas pressure.

$CO_2$ is added from the thermal press. We record pressure and temperature before and after the loading (see Supplementary Fig. 7 as an example). It permits to calculate very accurately the mole number of $CO_2$ introduced and so, we can estimate very accurately the loadings of $CO_2$.

The experimental method[36] is similar to the one used to calculate the solubility of $CO_2$ in alkanolamine amine solution at equilibrium. The method considered is based on the "static-synthetic method". More details concerning the method are presented in the Supplementary Methods.

During the absorption of the $CO_2$, we take samples to follow the evolution of the vapor composition (and so $CO_2$ partial pressure) as a function of time. When the equilibrium is reached (constant pressure and constant temperature), the vapor phase composition is determined.

We have used the GERG 2008 Equation of state[37] implemented in REFPROP 10.0[38] to estimate the densities of the vapor phase which is a mixture of $CO_2$ and $CH_4$.

The calculation of the acid gas solubility in the solvent is based on mass balance.

The volume of the liquid phase is obtained by considering the mole number of solvent introduced and its density at the temperature of measurement.

$$V^{L} = \frac{n_{solvent}}{\rho(T_{cell})} \quad (9)$$

Consequently, the volume of the vapor phase is calculated by difference between the total volume and the volume of the liquid phase.

$$V^{V} = V^{T} - V^{L} \quad (10)$$

If the introduction of the solute doesn't modify the level of the liquid interface in the equilibrium cell, we can consider Eq. (11).

$$V^{L} = \pi r_{cell}^{2} h_{liq} \quad (11)$$

Where $r_{cell}$ is the radius of the equilibrium cell, $h_{liq}$ the level of the vapor liquid interface.

The mole number of solute in the vapor phase is calculated by considering the density of the gas at the temperature and pressure of solute ($P_{Solute} = P_{cell} - P_{solvent}^{sat}$). REFPROP v10.0 is used to calculate this density $\rho^{V}(T_{cell}, P_{solute})$. In the case of a mixture, the global composition needs to be considered $\rho^{V}(T_{cell}, P_{solute}, y)$.

The volume of the vapor phase is used to calculate the mole number of solute in the vapor phase (Eq. (12)).

$$n^{V} = V^{V} \rho^{V}(T_{cell}, P_{solute}) \quad (12)$$

In the case of a mixture, the same equation is used to calculate the total mole number of solute in the vapor phase.

So, the mole number of solute in the liquid phase is determined by considering Eq. (13).

$$n^{L} = n^{T} - n^{V} \quad (13)$$

In the case of the mixture, the mole number of each species is calculated by considering the global composition of the mixture (z) and the composition of the vapor phase (y), Eq. (14).

$$n_{i}^{L} = z_{i} n^{T} - y_{i} n^{V} \quad (14)$$

The solubility is determined with Eq. (15).

$$x_{i} = \frac{n_{i}}{\sum n_{j}} \quad (15)$$

## Data availability

All the experimental data are available in Supplementary Materials and at https://github.com/AxelRolov/CO2_chemical_solvents. Structures of compounds, descriptors and predicted values are also available at https://github.com/AxelRolov/CO2_chemical_solvents. The data are also deposited into a DOI-minting repository ZENODO: https://doi.org/10.5281/zenodo.6010667.

## Code availability

Jupyter notebooks containing the Python code used for model building, evaluation and virtual screening are available at https://github.com/AxelRolov/CO2_chemical_solvents. The code is also deposited into a DOI-minting repository ZENODO: https://doi.org/10.5281/zenodo.6010667. Python libraries used for machine-learning and OPERA software are freely available. ISIDA-Fragmentor is available upon reasonable request to Prof. Alexandre Varnek.

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

## Acknowledgements
This work was supported by the Carbon Capture Utilization and Storage (CCUS) transverse R&D program from TotalEnergies S.E.

## Author contributions
A.A.O. performed machine learning, analyzed, interpreted the data, and contributed to the writing of the manuscript. X.R. and E.W. performed the molecular simulations. A.Valtz and C.C. performed the experimental part of the work. G.M. and D.H. contributed to the machine-learning models. B.P. contributed to the planning of the research. A. Varnek conceived, planned, and guided the part of the research related to building machine-learning models. F.D.M. conceived, planned, guided the research, analyzed, and interpreted the data, and wrote the manuscript. All authors critically analyzed data, edited, and approved the manuscript.

## Competing interests
The authors declare no competing interests.
