## [Peer Review File · Communications Chemistry]

Reviewers' comments:

Reviewer #1 (Remarks to the Author):

This work presented on rapid screening of solvents for CO₂ capture is well written and presented. I believe it is work that would be worthy of publishing if a few small additions could be made, listed below in no particular order:

- Since molecular simulations play a central role in this work, the details of those simulations need to be included somewhere in the paper. Neither the manuscript nor the SI provide any of the normally required details for reproducibility: force field parameters, software package used, time steps, equilibration/production time steps, ensembles (NVT vs. NVE), choice of thermostat, etc. I know that the authors reference another paper for some of these details, but I think it's important to include at least in the SI of *this* paper.

- I question the claim, put forth in Figure 1, that the molecular simulation models are of "high precision" - how is this claim justified? It may be a difference of assumptions between different communities, but in many circles one could only achieve high precision from molecular modelling results if one is using ab initio levels of theory - particularly so when reactive species are involved. For what it's worth, I don't think the authors need "high precision" when they are doing high throughput screening - it only matters that the right solvent candidates are ultimately discovered. I think the manuscript should elaborate on the expected accuracy of these molecular simulation models.

Overall, I think this valuable work and look forward to it being published.

Reviewer #2 (Remarks to the Author):

In this work, Orlov et al. proposed a new method that combines molecular simulation and machine learning algorithms for multi-objective tasks. The novel method is used to design effective solvents for CO₂ capture and is experimentally validated. This original work offers a novel way for large-scale screening. The descriptions of all graphs are clear, and conclusions seem reliable. It is an interesting work, I would like to recommend this work for publication after the authors properly address the following questions proposed.

- 1) In this work, the authors screened a dataset to find the high-performance solvent. I suggest the authors choose a few compounds in the screening library and predict their properties such as absorption rates, Gibbs free energy by MD to further check the accuracy of the QSPR algorithm.
- 2) In this work, a dataset containing 124 samples was used, is it large enough? Could the authors explain how to choose the size of the training set?
- 3) Could the author introduce how to select the features for the machine learning algorithm?
- 4) I recommend the authors give some references about the relation between absorption rate and Gibbs free energy on the last paragraph of page 6.

Reviewer #3 (Remarks to the Author):

This manuscript reports a judicious combination of models, from DFT calculations to molecular dynamics simulations, to quantitative structure-property relations, furthermore complemented by some experimental tests, aimed at identifying better amine structures for the capture of CO₂ at low pressure.

After validating the models with the help of an experimental dataset, many new structures were investigated based on reasonable criteria. The results are promising, with some new amines that show improved performance when compared to the industry standard (although less so when a piperazine "adjuvant" is added).

The methods are well documented and the SI provides relevant information.

I consider this to be a significant contribution, suitable for publication.

The review of the state of the art could be much improved by citing other works of searching and screening amine structures. Just a short search showed articles on this strategy (e.g. [10.1016/j.egypro.2014.11.190](https://doi.org/10.1016/j.egypro.2014.11.190))

Concerning the form, the text requires many minor corrections on typesetting and the use of English. These are annotated in the review manuscript pdf file.

A novel methodology for the design of effective solvents for CO₂ capture

Alexey A. Orlov[§], Alain Valtz[‡], Christophe Coquelet[‡], Xavier Rozanska[°], Erich Wimmer[°], Gilles Marcou[§], Dragos Horvath[§], Bénédicte Poulain[†], Alexandre Varnek^{§}, Frédérick de Meyer^{†‡*}.*

§ Laboratory of Chemoinformatics, Faculty of Chemistry, University of Strasbourg, Strasbourg, 67081 France

‡ MINES ParisTech, PSL University, Centre of Thermodynamics of Processes (CTP), 35 rue St Honoré 77300 Fontainebleau, France

° Materials Design SARL, 42 avenue Verdier, 92120 Montrouge, France

† TOTALenergies S.E, OneTech, Gas & Low Carbon Entity, CCUS R&D Program, 2 Place Jean Millier, 92078 Paris, France

*corresponding author: frederick.de-meyer@totalenergies.com

Keywords: greenhouse gas, carbon dioxide, chemical solvents, tertiary amines, molecular simulations, chemoinformatics, machine learning

Summary

Carbon capture and storage technologies are projected to increasingly contribute to cleaner energy transitions by significantly reducing CO₂ emissions from fossil fuel-driven power and industrial plants.^{1,2} The **industry standard** **state-of-the-art** technology for CO₂ capture is the chemical absorption with aqueous alkanolamines^{3,4}. The important equipment cost and parasitic energy due to the thermal regeneration of the solvent prevents its widespread application. Unfortunately, solvents showing a lower regeneration energy tend to absorb CO₂ more slowly, resulting in much larger equipment. Therefore, new industrial solvents are often a mixture of amines with relatively low regeneration energy (e.g. tertiary amines) and an activator, piperazine, which increases the overall CO₂ absorption rate⁵. The search for new tertiary amines with better kinetics is thus of paramount importance. Improving the efficiency of experimental screening using computational tools is challenging due to the complex nature of chemical absorption. We have developed a novel computational approach that combines kinetic experiments, molecular simulations and machine learning for the *in silico* screening of hundreds of prospective candidates and identify a class of tertiary amines that absorbs CO₂ faster than a typical commercial solvent when mixed with piperazine, which was confirmed experimentally.

Main

Numerous technologies exist for capturing CO₂ including chemical absorption, cryogenic separation, removal with membranes and adsorption with zeolites or metal-organic frameworks.^{3,6} The cyclic chemical absorption and regeneration process based on common primary and secondary amines such as monoethanol amine (MEA) and diethanol amine (DEA) is the most mature in industrial applications^{3,5}. Unhindered primary and secondary amines react rapidly with CO₂ to form very stable carbamates. The amount of energy required for the regeneration of these solvents is large. Carbon capture applied to a coal-fired power plant may

reduce the net output of the plant by 30%⁶. With sterically hindered amines or tertiary amines like the standard methyldiethanolamine (MDEA), CO₂ is captured as bicarbonate, which has a much smaller heat of reaction than carbamate formation, resulting in regeneration energy savings⁷. Moreover, their CO₂ absorption capacity is much higher. Tertiary amines are therefore increasingly used in the high-pressure natural gas treatment industry to remove acid gases like CO₂. However, in general, the rate of direct bicarbonate formation is much lower than that of carbamate formation resulting in much slower CO₂ absorption rates with tertiary amines and thus in unacceptable large equipment for low pressure, anthropogenic (flue gas), CO₂ capture applications^{5,7}. To tackle this problem, several approaches were suggested. Several studies reported that the usage of a catalyst allows one to speed up the absorption of CO₂ and/or to lower the energetic cost of the solvent regeneration⁸. Another option, which is currently followed by the industry, consists in adding an activator, piperazine, significantly boosting the overall CO₂ absorption rate without increasing the regeneration energy too much⁹. A more straightforward strategy would be the identification of new tertiary amines with much higher absorption rates with respect to standard MDEA and to which piperazine can eventually be added. Since experimental measurement of CO₂ absorption kinetics is a time and labor-intensive process, the rational approach to the design of tertiary amines that can rapidly absorb CO₂ requires a quantitative model enabling to select only the best candidates for experimental measurements.

For the adsorption-based CO₂ capture process (which is still under development) a lower theoretical energy consumption is expected due to the weaker physical adsorption. Molecular simulations and machine learning have already been extensively used to perform virtual screening of hundreds of thousands of structures to identify potentially better materials for CO₂ adsorption.^{10,11} Until now it was not possible to apply a similar methodology for amines, because of the difficulty related to the computation of chemical reactions. Previously, machine learning algorithms were tentatively applied for modeling quantitative structure-property relationship (QSPR) of alkanolamines' CO₂ absorption-related properties¹²⁻¹⁷. However, the availability of

Specify from the outset that the first part of this paragraph refers to adsorption in solid materials. The words "the absorption-based CO₂ capture process" seem to mention only one specific process... maybe rephrase to "Concerning alternative processes based on adsorption in porous solids (still under development), ... "

only a very small amount of data points limited the applicability domain of the models. Hence, to address this issue, we developed and applied a novel methodology for the identification of new tertiary amines effectively absorbing CO₂ based on the combination of molecular simulations¹⁸ and machine learning. In parallel, a new experimental set-up for the measurement of CO₂ absorption rates has been specifically designed and put in place to validate the approach.

The workflow of the methodology is presented in Figure 1. Chowdhury et al.¹⁹ published a consistent experimental dataset of the absorption rates of CO₂ for 24 aqueous tertiary amines (313 K, 30 wt% amine). In the absence of a clear relationship between the structure or the chemical properties (e.g. the basicity) of the amines and the CO₂ absorption rates, we developed a molecular dynamics (MD) based model that can accurately predict those experimental CO₂ absorption rates.¹⁸ It was found that, while the basicity of the amine (quantified by the pKa) is important, the key to the precision of molecular simulations is the inclusion of subtle but important solvation effects in the calculation of the activation Gibbs free energy of the reaction with an accuracy better than 1 kJmol⁻¹. One of the important features of the MD model¹⁸ is the robustness to reasonable changes in the concentration of amine and temperatures, enabling to apply it to a rather wide range of experimental setups. Hence, the model was applied to predict the rates at 13 mol% of amines and at 323 K, because these conditions are more representative of industrial absorption⁵.

Being much less resource- and cost-demanding, molecular simulations can thus be used instead of the experiments to get enough data for building a reliable QSPR model with a wide applicability domain.

Figure 1. A workflow of the methodology suggested in this paper. (1) A high precision molecular simulations-based model for absorption rate prediction is developed¹⁸ and validated with experimental data on CO₂ absorption rates for 24 tertiary amines¹⁹. (2) This model is applied to a diverse dataset containing 100 tertiary amine structures to calculate the CO₂ absorption rate (R_{MD}) and free energy of absorption (ΔG_{MD}) (see Methods). (3) QSPR models were built for R_{MD} and ΔG_{MD} . (4) QSPR models were used to select perspective commercially available amines from public datasets. (5) Experimental measurement of CO₂ absorption rates for selected amines. (6) The most selective ones can be further studied and eventually tested in a pilot unit.

Thus, a dataset containing 100 structurally diverse tertiary amines was composed based on the in-house TotalEnergies’s dataset of amines with known experimental properties, complemented with the tertiary amines extracted from literature and public databases (PubChem²⁰, ZINC²¹). The selected compounds comprise diverse chemotypes, including linear and cyclic amines, diamines, amines containing thiol and thioether groups. Molecular simulations (see Methods) were performed for the initial set of 24 amines and for the selected set of 100 amines at 323 K and using a 13 mol% concentration of amine. From MD simulations absorption rates (R_{MD}) and free energies of absorption (ΔG_{MD}) were obtained. Notably, the R_{MD} values calculated at 313 and 323 K are highly correlated (Figure 2a, Spearman rank correlation coefficient (ρ) 0.99).

Figure 2. **a**, The absorption rates (R_{MD}) at 313 K (grey) and 323 K (orange) predicted using MD and the experimental absorption rates (R_{lit}) at 313 K. **b**, R_{MD} vs predicted pK_a values (pK_a). **c**, energy of absorption (ΔG_{MD}) predicted by MD vs R_{MD} . **d**, ΔG_{MD} vs predicted pK_a . The 24 amines from Chowdhury et al.^{18,19} are shown in orange. The 100 amines for which MD simulations were performed are shown in black. Industrially used reference compound (MDEA) is shown in green.

As shown in Figure 2b, the most rapidly absorbing compound according to the MD calculations and the data from Chowdhury et al.¹⁹ was 3-(Diethylamino)-1,2-propanediol (DEA-1,2-PD). However, most of the other compounds with the largest predicted rates of absorption (R_{MD}) contained either piperidine or pyrrolidine cycles. ^{This} It is in line with the data from Chowdhury et al.¹⁹, who showed that 3-piperidino-1,2-propanediol (3PP-1,2-PD) and 1-methyl-2-piperidineethanol were significantly faster than the industrially used methyldiethanolamine (MDEA). Figure 2c illustrates that the computed CO_2 absorption Gibbs free energies ΔG_{MD} are almost perfectly correlated with the CO_2 absorption rates, R_{MD} (Spearman ρ -0.98): the slower the CO_2 absorption, the higher the absorption Gibbs free energy. The correlation is not linear, and the decrease of ΔG_{MD} slows down significantly at higher CO_2 absorption rates.

Subsequently, machine learning algorithms were applied to establish quantitative structure-property relationships and screen a dataset of tertiary amines from a public dataset. The values of pK_a predicted by the OPERA model²² can be used as a rather good predictor for ΔG_{MD} . Indeed, the fitting of linear regression with the pK_a values as the only predictor leads to a reasonable predictive performance in cross-validation (Table S2). For modeling both end-points (ΔG_{MD} and R_{MD}), we implemented a machine learning workflow combining several machine learning algorithms and various descriptors of molecular structures. Thus, predicted pK_a values were complemented with other descriptor types: physico-chemical descriptors from OPERA and various types of ISIDA fragments.^{23,24} Finally, we chose a consensus model consisting of several random forest²⁵ and XGBoost²⁶ models built on a merged subset of ISIDA fragments and descriptors predicted by OPERA model. Although, the predictive accuracy in terms of RMSE is the same order of magnitude as in Kuenemann et al.¹² for absorption rates (Table S2, Figure S1), the applicability domain of our models is much larger, since the training set contained three times more compounds. It is worth noting that a QSPR model which did not allow one to achieve an excellent accuracy can still be useful for ranking the amines from the large compounds databases^{12,27}. Therefore, we retrieved from the public database ZINC²¹ the tertiary amines which were not too large ($M_w \leq 250 \text{ g mol}^{-1}$), not too lipophilic ($-1 \leq \text{clogP} \leq 1$), and readily available from suppliers. In total, more than 800 amines were screened virtually. Numerous amines outranking MDEA in terms of the predicted absorption rates (R_{QSPR}) were identified (Figure 3a). For example, various substituted piperidines were among the compounds with the largest R_{QSPR} (Figure 3a).

Figure 3. **a**, The absorption rates predicted by the QSPR model (R_{QSPR}) vs predicted pK_a values. **b**, R_{QSPR} vs experimentally measured absorption rate (ΔCO_2). **c**, ΔCO_2 vs predicted pK_a . **d**, free energies of absorption predicted by QSPR model (ΔG_{QSPR}) vs predicted pK_a . **e**, ΔG_{QSPR} vs ΔCO_2 . **f**, ΔG_{QSPR} vs R_{QSPR} . Amines present in the initial dataset from Chowdhury et al.^{18,19} are shown in orange. Amines selected for MD simulations in the present work are shown in black. The industrially used reference compound (MDEA) is shown in green. 8 novel amines which were not present in the training set are shown in blue. The CAS numbers of the most perspective compounds are shown.

Delta CO2 is not a good notation for a kinetic rate. $r(CO_2)$ or something similar would be preferable

In parallel, an experimental set-up was put in place to measure and compare the rate of CO_2 absorption in aqueous tertiary amines. For each experiment, the same initial amount of CO_2 was set in contact with the solvent and the evolution towards equilibrium of the partial pressure of CO_2 in the gas phase was measured over time. The slope of the absorption curve at the time at which 50% of the CO_2 was absorbed (with respect to the equilibrium value) was calculated (ΔCO_2). It is a measure of the rate of CO_2 absorption. Eighteen amines comprising 7 amines from the initial set of 24 amines from Chowdhury et al.¹⁹, 3 amines from the diverse dataset of 100 amines and 8 novel amines that were never present in the training set were purchased and an assessment of their absorption rate was performed (Figure 3b-c, e, Table S3). Both ΔG_{QSPR} and absorption rates R_{QSPR} were highly correlated with ΔCO_2 for 8 novel amines (Spearman ρ 0.93)

as well as the predicted pKa values. Five out of 8 purchased amines absorbed CO₂ faster than MDEA. Two amines: 1-Methyl- and 1-Ethyl-3-pyrrolidinol (EPOL) were especially effective. These compounds represent an interesting class of the tertiary amines, which to our knowledge has not been explored yet.

While tertiary amines like the standard MDEA are often used for high pressure natural gas treatment, they are not suitable for low pressure anthropogenic CO₂ removal due to the low CO₂ absorption rate. Activators such as piperazine can be added to enhance the CO₂ absorption rate. The impact of piperazine is shown in Figure 4 for two amines, namely MDEA and EPOL. The latter is a tertiary amine that has been selected for its fast CO₂ absorption rate following the virtual screening. In the absence of piperazine EPOL absorbs CO₂ much faster than MDEA. Addition of piperazine significantly enhances the CO₂ absorption rates with EPOL + PZ showing the fastest absorption.

Figure 4. Experimentally measured CO₂ absorption rate (ΔCO_2) of standard MDEA and EPOL, a new amine suggested in this work, and their mixtures with piperazine (+PZ). Aqueous alkanolamine mixtures contain 13 mol% amine and water and mixtures with PZ contain 11 mol% amine, 2.5 mol% PZ and water.

In conclusion, a novel methodology for computer-aided design of new tertiary amines effectively absorbing CO₂ was suggested in this paper. The methodology is based on the combination of state-of-the-art molecular dynamics simulations that generate a sufficiently large dataset that are used as an input for machine learning modelling followed by large scale virtual screening. In parallel, the approach is experimentally validated. It allowed the identification of amines that absorb CO₂ faster than those currently used in the industry. Since the development of an optimal solvent is a multi-objective task, we believe that the proposed methodology can be provisionally repurposed to application for modeling of other industrially important properties of alkanolamine based solvents.

METHODS

Molecular simulations

The approach developed recently and described in Rozanska *et al.* was used to compute the rates¹⁸ of CO₂ absorption in aqueous amine solvents, which relies primarily on the solvation properties of OH⁻, CO₂, and HCO₃⁻. In this model the tertiary amine solely acts as a base.

$$R_{\text{MD}} = A(T) \times \exp\left(\frac{-\Delta G^\ddagger}{RT}\right) \times [\text{CO}_2][\text{OH}^-] \quad (1)$$

The rates are obtained from ~~the~~ Eq. (1) where R_{MD} is the absorption rate, $[\text{CO}_2]$ and $[\text{OH}^-]$ are the concentrations of carbon dioxide molecules and hydroxyl anions, respectively, ΔG^\ddagger is the Gibbs free energy barrier of the reaction $\text{CO}_2 + \text{OH}^- \rightarrow \text{HCO}_3^-$, RT is the macroscopic thermodynamic energy unit, where R is the universal gas constant and T the absolute temperature, and $A(T)$ is a temperature-dependent pre-exponential factor. In equation (1), ΔG^\ddagger is

obtained from a Polanyi-Evans relation with as input the energy differences of solvation of $\text{OH}^- + \text{CO}_2$ (reactants) and HCO_3^- (product) computed in the 124 aqueous amine solvents. The concentrations $[\text{CO}_2][\text{OH}^-]$ are obtained numerically solving pH equations, and $A(T)$ is fitted using the experimental rates of the reaction $\text{CO}_2 + \text{OH}^-$ in 10 aqueous amine solvents. The Polanyi-Evans relation between ΔG^\ddagger and energy differences of solvation, ΔG , of $\text{OH}^- + \text{CO}_2$ and HCO_3^- is given by Eq. (2).

$$\Delta G^\ddagger = a \Delta G(T) + b \quad (2)$$

where a and b are fitted to reproduce the experimental rates in pure water and 10 aqueous amine solvents and $\Delta G(T)$ is the energy difference of solvation of $\text{OH}^- + \text{CO}_2$ and HCO_3^- obtained from molecular dynamics simulations. Additional details and the values for $A(T)$, a , and b can be found in Rozanska *et al.*¹⁸

For the calculation of the regeneration energy, the following 3 reactions are considered:

italic for quantities

The free energy of absorption is ΔG_4 ($= \Delta G_{\text{MD}}$ in Figure 2) $= \Delta G_3 + \Delta G_5$ with ΔG_3 calculated from the molecular simulations ($\Delta G(T)$ in Eq. (2)) in every aqueous amine and ΔG_5 calculated from the amine pK_a .

Quantitative structure-property relationship modeling

All compound structures were standardized using RDKit²⁸ nodes in KNIME.²⁹ The standardization procedure included aromatization, stereochemistry depletion, removal of salts/solvents, neutralization, removal of explicit hydrogens. Standardized structures for 124

amines are given in the Table S1 and at the https://github.com/AxelRolov/CO2_chemical_solvents.

193 different ISIDA fragment descriptors were generated using the Fragmentor17 software.^{23,24} These fragments represent either sequences (the shortest topological paths with an explicit presentation of all atoms and bonds), atom pairs, or triplets (all the possible combinations of 3 atoms in a graph with the topological distance between each pair indicated).

Various physico-chemical properties (pKa, logP, melting and boiling points, vapour pressure, water solubility, etc.) and several substructural fragments counts (ring count, heavy atom count, etc.) used as descriptors, were calculated using OPERA v.2.6²².

All descriptors used in this work are available at https://github.com/AxelRolov/CO2_chemical_solvents.

Prior to the application of machine learning algorithms R_{MD} and ΔG_{MD} values were transformed to a logarithmic scale, i.e., ^{the} negative value of decimal logarithm was taken ($-\log_{10}R_{MD}$, $-\log_{10}(-\Delta G_{MD})$).

Random forest (RF): RF algorithm²⁵ implemented in sci-kit learn library (v. 0.22.1)³⁰, was used. The following hyperparameters were optimized (grid search): number of trees (100, 300, 1000), number of features (all features, one-third of all features, \log_2 of the number of features), the maximum depth of the tree (10, 30, full tree), bootstrapping (with and without the usage of bootstrap samples for building the tree).

XGBoost (XGB): XGBoost algorithm²⁶ as implemented in XGBoost python module (v.1.2.0)³¹ was used. The following hyperparameters were tuned during optimization (grid search): number of trees (50, 100, 300, 500), number of features (all features, 70% of all features), number of samples (all samples, 70% of all samples), the maximum depth of the tree (5, 20, full tree), learning rate (0.3, 0.1, 0.5, 0.05). All other parameters were left as default.

Support vector regression (SVR): SVR algorithm³² implemented in sci-kit learn library (v. 0.22.1), was used. The descriptors were scaled to the [0,1] range before applying the algorithm. The following hyperparameters were tuning during optimization (grid search): kernel (linear, rbf, poly, sigmoid), kernel coefficient (1, 0.1, 0.01, 0.001, 0.0001), regularization parameter (0.1, 1, 10, 100, 1000).

The modeling workflow was implemented using the sci-kit learn library (v. 0.22.1) in Python 3.7 scripting language (Figure S2). Identical modeling workflows were used for modeling absorption rates (R_{MD}) and energies of absorption (ΔG_{MD}). The values were expressed as negative logarithms of base 10. At the first stage of the modeling, a machine learning algorithm: RF, SVR and XGB were tested in 5-fold cross-validation, which was repeated 5 times. For each descriptor set, the model's measures of performance were calculated and several models with a coefficient of determination $Q^2_{CV} \geq 0.6$ for (R_{MD}) and $Q^2_{CV} \geq 0.7$ for (ΔG_{MD}) were selected for consensus modeling. Consensus models were built for each descriptor type separately. In order to assess a propensity to predict data never seen during the training of the model, a nested cross-validation procedure³³ has been implemented. Here the method hyperparameters were found by optimizing the model performance in the 5-fold cross-validation inner loop, while prediction was made for the test set from the outer loop, which represent a fold of outer 5-fold cross-validation cycle. In order to avoid a bias with the compounds numbering in the parent set, this procedure was repeated 5 times after reshuffling of the compounds. In such a way, the overall performance of the model (Q^2_{NCV} , $RMSE_{NCV}$, MAE_{NCV}) were estimated as an average of related statistical parameters obtained for each (out of 5) individual cross-validation loop.

The Eqs. (6 to 8) were used to calculate the measures of the model's performance in cross-validation:

$$Q^2_{CV} = \frac{\sum_{j=1}^5 \left(1 - \frac{\sum_{i=1}^n (y_{i,exp} - y_{i,pred})^2}{\sum_{i=1}^n (y_{i,exp} - \bar{y})^2} \right)}{5} \quad (6)$$

$$RMSE_{CV} = \frac{\sum_{j=1}^5 \sqrt{\frac{\sum_{i=1}^n (y_{i,exp} - y_{i,pred})^2}{n}}}{5} \quad (7)$$

$$MAE_{CV} = \frac{\sum_{j=1}^5 \sum_{i=1}^n \frac{|y_{i,exp} - y_{i,pred}|}{n}}{5} \quad (8)$$

Above, n is the number of compounds in the learning set, $y_{i,exp}$, $y_{i,pred}$ experimental and values predicted in 5-fold cross-validation for compound i from the learning set, j is the index of the repetition of the 10-fold cross-validation procedure.

Each of the selected models was then associated with an Applicability Domain (AD), defined as a boundary box. The pool of selected models extracted from the given data set can now be used as a consensus predictor, returning for each input solvent candidate a mean value of solubility estimates and its standard deviation, taken over the predictions returned by each model in the pool or, alternatively, over the predictions returned by only those models having the candidate within their AD.

Outlying data points were defined as the data points, for which absolute errors ($|y_{exp} - y_{pred}|$) from cross-validation were larger than $2 \times RMSE_{CV}$ threshold.

The absence of chance correlation was checked through the Y-randomization procedure. A Y-randomization test was performed in the following way: $-\log_{10}\chi$ values (y values) were shuffled, models were built using shuffled values and the values from the corresponding cross-validation test set were calculated. This procedure was repeated 100 times for each fold and the maximum values of the out-of-bag coefficient of determination were reported.

A library for virtual screening was performed in the following way. At first, all compounds from ZINC database with molecular weight no larger than 250 g/mol and calculated $\log P$ in the range of (-1,1) were retrieved. Structures were standardized and then filtered. All compounds which did not contain tertiary amines, compounds, containing double bonds, aromatic rings,

primary or secondary amine groups, ketones and sulfur containing compounds except for thiols and thioethers were removed. Structures of screened compounds as well as predicted values are available at https://github.com/AxelRolov/CO2_chemical_solvents.

Experimental measurement of CO₂ absorption rates

To measure the kinetics of absorption and desorption of acid gases in aqueous amine solutions, a thermo regulated constant interfacial area Lewis type reactor cell was used³³. The reactor (Figure S1) is equipped with an internal stirring system (magnetic stirrer) with external motor. The operator has to take care to select the speed of stirring without disturbing the interface (interface must be flat). Pictures of the equipment are given in Supplementary Information (Figure S4). Temperature is given by two platinum probes located at the upper and lower flanges (with possibility to determine the gradient of temperature). The cell is immersed in a liquid bath. An electric resistor is introduced into the upper flange in order to control the gradient of temperature and avoid condensation of water and amine. Two capillary samplers are adapted to sample the vapor phase. The capillary samplers (ROLSI®, Armines' patent) are capable of withdrawing and sending micro samples to a gas chromatograph without perturbing the equilibrium conditions over numerous samplings, thus leading to repeatable and reliable results. Analytical work was carried out using a gas chromatograph (PERICHROM model PR2100, France) equipped with a thermal conductivity detector (TCD) connected to a data software system. Helium is used as the carrier gas in this experiment. The model of the GC column is Porapak R (Porapak R 80 / 100 mesh, 1 m x 2 mm ID Silcosteel). Each ROLSI® sampler is connected to a TCD. A tube allows either to evacuate or to introduce CO₂ from or into the cell. The kinetics of gas absorption is determined by recording the pressure drop through a calibrated pressure transducer. A computer equipped with data acquisition system records the pressure as a function of time.

The experimental procedure is the following:

The desired amount of solvent is introduced into the cell. The density obtained using a low-pressure vibrating tube densitometer (Anton Paar DSA 5000) is used to determine the exact mole number of solvent.

At least 5 bar of methane is added. We add methane because with this configuration, it is not possible to sample at pressures lower than GC carrier gas pressure.

CO₂ is added from the thermal press. We record pressure and temperature before and after the loading (See Figure S5 as an example). It permits to calculate very accurately the mole number of CO₂ introduced and so, we can estimate very accurately the loadings of CO₂.

The experimental method³⁴ is similar to the one used to calculate the solubility of CO₂ in alkanolamine amine solution at equilibrium. The method considered is based on the “static-synthetic method”. More details concerning the method are presented in the Supplementary Information.

During the absorption of the CO₂, we take samples to follow the evolution of the vapor composition (and so CO₂ partial pressure) as a function of time. When the equilibrium is reached (constant pressure and constant temperature), the vapor phase composition is determined.

We have used the GERG 2008 Equation of state³⁵ implemented in REFPROP 10.0³⁶ to estimate the densities of the vapor phase which is a mixture of CO₂ and CH₄.

The calculation the acid gas solubility in the solvent is based on mass balance.

The volume of liquid phase is obtained by considering the mole number of solvent introduced and its density at the temperature of measurement.

$$V^L = \frac{n_{\text{solvent}}}{\rho(T_{\text{cell}})} \quad (9)$$

Consequently, the volume of the vapor phase is calculated by difference from the total volume and the volume of the liquid phase.

$$V^V = V^T - V^L \quad (10)$$

If the introduction of the solute doesn't modify the level of the liquid interface in the equilibrium cell, we can consider equation 11.

$$V^L = \pi r_{cell}^2 h_{liq} \quad (11)$$

Where r_{cell} is the radius of the equilibrium cell, h_{liq} the level of the vapor liquid interface.

The mole number of solute in the vapor phase is calculated by considering the density of the gas at the temperature and pressure of solute ($P_{solute} = P_{cell} - P_{solvent}^{sat}$). REFPROP v10.0 is used to calculate this density $\rho^V(T_{cell}, P_{solute})$. In case of mixture, the global composition have to be considered $\rho^V(T_{cell}, P_{solute}, y)$.

The volume of the vapor phase is used to calculate the mole number of solute in the vapor phase (Eq. 12).

$$n^V = V^V \rho^V(T_{cell}, P_{solute}) \quad (12)$$

In case of mixture, the same equation is used to calculate the total mole number of solute in the vapor phase.

So, the mole number of solute in the liquid phase is determined by considering Eq. 13.

$$n^L = n^T - n^V \quad (13)$$

In case of mixture, the mole number of each species is calculated by considering the global composition of the mixture (z) and the composition of the vapor phase (y), Eq. 14.

$$n_i^L = z_i n^T - y_i n^V \quad (14)$$

The solubility is determined with Eq. 15.

$$x_i = \frac{n_i}{\sum n_j} \quad (15)$$

The experimental points are given in the supplementary information.

SUPPLEMENTARY INFORMATION.

This file contains Supplementary Methods, Tables S1–S5, Figs. S1–S5 and references.

AUTHOR INFORMATION

Affiliations

1. Laboratory of Chemoinformatics, Faculty of Chemistry, University of Strasbourg, Strasbourg, 67081 France
2. MINES ParisTech, PSL University, Centre of Thermodynamics of Processes (CTP), 35 rue St Honoré 77300 Fontainebleau, France
3. Materials Design SARL, 42 avenue Verdier, 92120 Montrouge, France
4. TOTALEnergies S.E, OneTech, Gas & Low Carbon Entity, CCUS R&D Program, 2 Place Jean Millier, 92078 Paris, France

Contributions

A.A.O. performed machine learning, analyzed, interpreted the data, and contributed to writing of the manuscript. X.R. and E.W. performed the molecular simulations. A. Valtz and C.C. performed the experimental part of the work. G.M. and D.H. contributed to the machine learning models. B.P. contributed to the planning of the research. A. Varnek conceived, planned, and guided the part of the research related to building machine learning models. F.D.M. conceived, planned, guided the research, analyzed and interpreted the data and wrote the manuscript. All authors critically analyzed data, edited and approved the manuscript.

CORRESPONDING AUTHORS

Correspondence to Frédérick de Meyer and Alexandre Varnek.

ETHICS DECLARATIONS

Competing interests

The authors declare no competing interests.

DATA AVAILABILITY

All the experimental data is available in Supplementary Materials and at https://github.com/AxelRolov/CO2_chemical_solvents. Structures of compounds, descriptors and predicted values are also available at https://github.com/AxelRolov/CO2_chemical_solvents.

CODE AVAILABILITY

Jupyter notebooks containing the Python code used for model building, evaluation and virtual screening are available at https://github.com/AxelRolov/CO2_chemical_solvents. Python libraries used for machine learning and OPERA software are freely available. ISIDA-Fragmentor is available upon reasonable request to Prof. Alexandre Varnek.

REFERENCES

1. Net Zero by 2050 – Analysis. *IEA* <https://www.iea.org/reports/net-zero-by-2050>.
2. Hepburn, C. *et al.* The technological and economic prospects for CO₂ utilization and removal. *Nature* **575**, 87–97 (2019).
3. Bui, M. *et al.* Carbon capture and storage (CCS): the way forward. *Energy Environ. Sci.* **11**, 1062–1176 (2018).
4. Rochelle, G. T. Amine Scrubbing for CO₂ Capture. *Science* **325**, 1652–1654 (2009).
5. Brickett Lynn. *Carbon Dioxide Capture Handbook*. <https://netl.doe.gov/sites/default/files/netl-file/Carbon-Dioxide-Capture-Handbook-2015.pdf>. (2015).
6. Smit, B. Carbon Capture and Storage: introductory lecture. *Faraday Discuss.* **192**, 9–25 (2016).
7. N.Borhani, T. & Wang, M. Role of solvents in CO₂ capture processes: The review of selection and design methods. *Renew. Sustain. Energy Rev.* **114**, 109299 (2019).

8. de Meyer, F. & Bignaud, C. The use of catalysis for faster CO₂ absorption and energy-efficient solvent regeneration: An industry-focused critical review. *Chem. Eng. J.* **428**, 131264 (2022).
9. Li, L. *et al.* Amine blends using concentrated piperazine. *Energy Procedia* **37**, 353–369 (2013).
10. Lin, L.-C. *et al.* In silico screening of carbon-capture materials. *Nat. Mater.* **11**, 633–641 (2012).
11. Boyd, P. G. *et al.* Data-driven design of metal–organic frameworks for wet flue gas CO₂ capture. *Nature* **576**, 253–256 (2019).
12. Kuenemann, M. A. & Fourches, D. Cheminformatics Modeling of Amine Solutions for Assessing their CO₂ Absorption Properties. *Mol. Inform.* **36**, 1600143 (2017).
13. Khareshi, S., Riahi, S., Mohammadi-Khanaposhtani, M. & shokrollahzadeh, H. Prediction of Amines Capacity for Carbon Dioxide Absorption Based on Structural Characteristics. *Ind. Eng. Chem. Res.* **58**, 8763–8771 (2019).
14. Rezaei, B., Riahi, S. & Gorji, A. E. Molecular investigation of amine performance in the carbon capture process: Least squares support vector machine approach. *Korean J. Chem. Eng.* **37**, 72–79 (2020).
15. Cheng, J. *et al.* Quantitative Relationship between CO₂ Absorption Capacity and Amine Water System: DFT, Statistical, and Experimental Study. *Ind. Eng. Chem. Res.* **58**, 13848–13857 (2019).
16. Gonfa, G., Bustam, M. A. & Shariff, A. M. Quantum-chemical-based quantitative structure-activity relationships for estimation of CO₂ absorption/desorption capacities of amine-based absorbents. *Int. J. Greenh. Gas Control* **49**, 372–378 (2016).
17. Porcheron, F. *et al.* Graph Machine Based-QSAR Approach for Modeling Thermodynamic Properties of Amines: Application to CO₂ Capture in Postcombustion. *Oil Gas Sci. Technol. – Rev. D’IFP Energ. Nouv.* **68**, 469–486 (2013).

18. Rozanska, X., Wimmer, E. & de Meyer, F. Quantitative Kinetic Model of CO₂ Absorption in Aqueous Tertiary Amine Solvents. *J. Chem. Inf. Model.* **61**, 1814–1824 (2021).
19. Chowdhury, F. A., Yamada, H., Higashii, T., Goto, K. & Onoda, M. CO₂ Capture by Tertiary Amine Absorbents: A Performance Comparison Study. *Ind. Eng. Chem. Res.* **52**, 8323–8331 (2013).
20. Kim, S. *et al.* PubChem in 2021: new data content and improved web interfaces. *Nucleic Acids Res.* **49**, D1388–D1395 (2021).
21. Sterling, T. & Irwin, J. J. ZINC 15 – Ligand Discovery for Everyone. *J. Chem. Inf. Model.* **55**, 2324–2337 (2015).
22. Mansouri, K., Grulke, C. M., Judson, R. S. & Williams, A. J. OPERA models for predicting physicochemical properties and environmental fate endpoints. *J. Cheminformatics* **10**, 10 (2018).
23. Varnek, A. *et al.* ISIDA - Platform for Virtual Screening Based on Fragment and Pharmacophoric Descriptors. *Curr. Comput. Aided-Drug Des.* **4**, 191–198 (2008).
24. Ruggiu, F., Marcou, G., Varnek, A. & Horvath, D. ISIDA Property-Labelled Fragment Descriptors. *Mol. Inform.* **29**, 855–868 (2010).
25. Breiman, L. Random Forests. *Mach. Learn.* **45**, 5–32 (2001).
26. Chen, T. & Guestrin, C. XGBoost: A Scalable Tree Boosting System. in *Proceedings of the 22nd ACM SIGKDD International Conference on Knowledge Discovery and Data Mining* 785–794 (ACM, 2016). doi:10.1145/2939672.2939785.
27. Tropsha, A., Gramatica, P. & Gombar, V. K. The Importance of Being Earnest: Validation is the Absolute Essential for Successful Application and Interpretation of QSPR Models. *QSAR Comb. Sci.* **22**, 69–77 (2003).
28. *RDKit: Open-source cheminformatics*; <http://www.rdkit.org>.
29. Open for Innovation. *KNIME* <https://www.knime.com/open-for-innovation-0>.

30. Pedregosa, F. *et al.* Scikit-learn: Machine Learning in Python. *J. Mach. Learn. Res.* **12**, 2825–2830 (2011).
31. https://xgboost.readthedocs.io/en/latest/python/python_intro.html (accessed 31.05.21).
32. Cortes, C. & Vapnik, V. Support-vector networks. *Mach. Learn.* **20**, 273–297 (1995).
33. Baumann, D. & Baumann, K. Reliable estimation of prediction errors for QSAR models under model uncertainty using double cross-validation. *J. Cheminformatics* **6**, 47 (2014).
34. Coquelet, C., Valtz, A. & Théveneau, P. *Experimental Determination of Thermophysical Properties of Working Fluids for ORC Applications. Organic Rankine Cycles for Waste Heat Recovery - Analysis and Applications* (IntechOpen, 2019). doi:10.5772/intechopen.87113.
35. Kunz, O. & Wagner, W. The GERG-2008 Wide-Range Equation of State for Natural Gases and Other Mixtures: An Expansion of GERG-2004. *J. Chem. Eng. Data* **57**, 3032–3091 (2012).
36. Eric Lemmon, Marcia Huber, & Mark McLinden. NIST Standard Reference Database 23: Reference Fluid Thermodynamic and Transport Properties-REFPROP, Version 9.1. (2013).

Reviewers' comments:

Reviewer #1 (Remarks to the Author):

This work presented on rapid screening of solvents for CO₂ capture is well written and presented. I believe it is work that would be worthy of publishing if a few small additions could be made, listed below in no particular order:

- Since molecular simulations play a central role in this work, the details of those simulations need to be included somewhere in the paper. Neither the manuscript nor the SI provide any of the normally required details for reproducibility: force field parameters, software package used, time steps, equilibration/production time steps, ensembles (NVT vs. NVE), choice of thermostat, etc. I know that the authors reference another paper for some of these details, but I think it's important to include at least in the SI of *this* paper.

Response: We now added in the Supporting Information a detailed section that describes the protocol and software packages that were used for the molecular dynamics simulations.

- I question the claim, put forth in Figure 1, that the molecular simulation models are of "high precision" - how is this claim justified? It may be a difference of assumptions between different communities, but in many circles one could only achieve high precision from molecular modelling results if one is using ab initio levels of theory - particularly so when reactive species are involved. For what it's worth, I don't think the authors need "high precision" when they are doing high throughput screening - it only matters that the right solvent candidates are ultimately discovered. I think the manuscript should elaborate on the expected accuracy of these molecular simulation models.

Response:

We have added the following sentence to the caption of Figure 1 to justify the wording "high precision": The accuracy of the Gibbs free energies of absorption is better than 1 kJ mol⁻¹ in comparison to experimental values¹⁸.

It is correct that it only matters that the right solvent candidates are ultimately discovered. For tertiary amines, the CO₂ absorption rates are mostly within the same order of magnitude (see Fig2b), so it is important to have a high accuracy of the Gibbs free energies.

Overall, I think this valuable work and look forward to it being published.

Reviewer #2 (Remarks to the Author):

In this work, Orlov et al. proposed a new method that combines molecular simulation and machine learning algorithms for multi-objective tasks. The novel method is used to design effective solvents for CO₂ capture and is experimentally validated. This original work offers a novel way for large-scale screening. The descriptions of all graphs are clear, and conclusions seem reliable. It is an interesting work, I would like to recommend this work for publication after the authors properly address the following questions proposed.

1) In this work, the authors screened a dataset to find the high-performance solvent. I suggest the authors choose a few compounds in the screening library and predict their properties such as absorption rates, Gibbs free energy by MD to further check the accuracy of the QSPR algorithm.

Response:

When building a QSPR model from a dataset obtained from MD, part of the dataset has been used for training (model building) and the other part has been used for validation. Different training and validation sets have been defined for cross-validation. This is the usual procedure to check the accuracy of the QSPR model.

The ultimate validation of any QSPR model is based on the comparison of predicted and experimentally measured values. In this context, we don't think that running more time consuming MD simulations is really needed. Instead, for 8 amines selected from the screening library we compared predicted values of Gibbs free energy (ΔG_{QSPR}) and absorption rate (R_{QSPR}) with experimentally measured absorption efficiency $r(\text{CO}_2)$. Excellent value of Spearman correlation coefficient (0.93) was observed.

2) In this work, a dataset containing 124 samples was used, is it large enough? Could the authors explain how to choose the size of the training set?

Response:

The a model performance depends not only on the training set size but also on the structural diversity of training objects. Our experience shows that for structurally homogeneous set of structures even several dozens might be sufficient.

A dataset of 124 amines used in this study is the largest set of CO₂ chemical absorbants reported in the literature so far. Its relevancy was demonstrated by reasonably high predictive performance of the models for absorption energy (Q₂>0.78 in cross-validation) built on this set and efficiency of these models in discovery of new absorbants.

Further increase of the dataset size won't necessarily improve the accuracy of predictions but will certainly enlarge the model's applicability domain and, in such a way, its efficiency in virtual screening.

3) Could the author introduce how to select the features for the machine learning algorithm?

Response:

Before applying a machine learning algorithm, one can perform supervised or unsupervised feature selection procedures in order to reduce the number of features (molecular descriptors) used for modeling. For example, one of the widely used approaches for unsupervised feature selection is removing correlated and low-variance features. However, the algorithms used in our work (random forest, extreme gradient boosting, support vector regression) do not necessarily require feature selection procedures and therefore, we did not use them in our modeling workflow.

4) I recommend the authors give some references about the relation between absorption rate and Gibbs free energy on the last paragraph of page 6.

Response: we have added a reference to the book “Reaction Rate Theory and Rare Events”, Chapter 22 (Free Energy Relationships) written by Baron Peters (Elsevier, 2017).

Reviewer #3 (Remarks to the Author):

This manuscript reports a judicious combination of models, from DFT calculations to molecular dynamics simulations, to quantitative structure-property relations, furthermore complemented by some experimental tests, aimed at identifying better amine structures for the capture of CO₂ at low pressure.

After validating the models with the help of an experimental dataset, many new structures were investigated based on reasonable criteria. The results are promising, with some new amines that show improved performance when compared to the industry standard (although less so when a piperazine "adjuvant" is added).

The methods are well documented and the SI provides relevant information.

I consider this to be a significant contribution, suitable for publication.

The review of the state of the art could be much improved by citing other works of searching and screening amine structures. Just a short search showed articles on this strategy (e.g. 10.1016/j.egypro.2014.11.190)

Response: We have added the following to the text:

Amines were rationally designed based on physical and thermodynamic properties and the CO₂ absorption rates were measured experimentally for only the most promising candidates^{12,13}.

Concerning the form, the text requires many minor corrections on typesetting and the use of English. These are annotated in the review manuscript pdf file.

Response : Thank you. We have implemented all corrections.

REVIEWERS' COMMENTS:

Reviewer #1 (Remarks to the Author):

The authors have addressed my minor concerns. Thank you.

Reviewer #2 (Remarks to the Author):

I accept the change of the manuscript made by the authors, and I recommend publication as the manuscript is now.

Reviewer #3 (Remarks to the Author):

The revised version incorporates additions and corrections that improve the clarity of the manuscript (including the molecular simulation details in the Supp Info). This reviewer's comments on the initial version were addressed in the new version.